# Hyperoxidized Peroxiredoxin 2 Is a Possible Biomarker for the Diagnosis of Obstructive Sleep Apnea

**DOI:** 10.3390/antiox11122486

**Published:** 2022-12-17

**Authors:** Shin Koike, Haruka Sudo, Satori Turudome, Masako Ueyama, Yoshiaki Tanaka, Hiroshi Kimura, Yo-Ichi Ishida, Yuki Ogasawara

**Affiliations:** 1Department of Analytical Biochemistry, Meiji Pharmaceutical University, 2-522-1 Noshio, Kiyose, Tokyo 204-8588, Japan; 2Laboratory of Biochemistry, Department of Clinical Pharmacy, Faculty of Pharmaceutical Sciences, Shonan University of Medical Sciences, 16-10 Kamishinano, Totsuka-ku, Yokohama, Kanagawa 244-0806, Japan; 3Sleep Apnea Syndrome Treatment Center, Fukujuji Hospital, Japan Anti-Tuberculosis Association, 3-1-24 Matsuyama, Kiyose, Tokyo 204-8522, Japan; 4Department of Pulmonary Medicine and Oncology, Graduate School of Medicine, Nippon Medical School, 1-1-5 Sendagi, Bunkyo-ku, Tokyo 113-8602, Japan; 5Department of Respiratory Medicine, Fukujuji Hospital, Japan Anti-Tuberculosis Association, 3-1-24 Matsuyama, Kiyose, Tokyo 204-8522, Japan

**Keywords:** peroxiredoxin 2, hyperoxidation, obstructive sleep apnea, biomarker, hypoxia, oxidative stress

## Abstract

Peroxiredoxin (Prx) 2 in red blood cells (RBCs) reacts with various reactive oxygen species and changes to hyperoxidized Prx2 (Prx2-SO_2/3_). Therefore, Prx2 may serve as an indicator of oxidative stress in vivo. This study aimed to analyze Prx2-SO_2/3_ levels in clinical samples to examine whether the oxidation state of Prx2 in human RBCs reflects the pathological condition of oxidative stress diseases. We first focused on obstructive sleep apnea (OSA), a hypoxic stress-induced disease of the respiratory system, and investigated the levels of Prx2-SO_2/3_ accumulated in the RBCs of OSA patients. In measurements on a small number of OSA patients and healthy subjects, levels of Prx2-SO_2/3_ accumulation in patients with OSA were clearly increased compared to those in healthy subjects. Hence, we proceeded to validate these findings with more samples collected from patients with OSA. The results revealed significantly higher levels of erythrocytic Prx2-SO_2/3_ in patients with OSA than in healthy subjects, as well as a positive correlation between the severity of OSA and Prx2-SO_2/3_ levels in the RBCs. Moreover, we performed a chromatographic study to show the structural changes of Prx2 due to hyperoxidation. Our findings demonstrated that the Prx2-SO_2/3_ molecules in RBCs from patients with OSA were considerably more hydrophilic than the reduced form of Prx2. These results implicate Prx2-SO_2/3_ as a promising candidate biomarker for OSA.

## 1. Introduction

Obstructive sleep apnea (OSA) is a chronic respiratory disorder that is observed in up to 50% of the population in certain countries, and which globally affects nearly one billion adults aged 30–69 years [1]. OSA is a common sleep-related breathing disorder characterized by clinical symptoms (e.g., daytime sleepiness) and at least five events per hour of narrowing (apnea or hypopnea) of the upper airway that impairs normal ventilation during sleep [2]. The risk of developing cardiovascular disorders, such as ischemic heart disease, heart failure, arrhythmia, stroke, and transient ischemic attack, is relatively high in patients with OSA [3,4]. Full-montage polysomnography is still considered the gold standard for diagnosing OSA. This technique is complex, expensive, causes discomfort to patients, and requires a dedicated sleep laboratory with highly qualified personnel [5]. Consequently, clinical guidelines highlight the need for biomarkers to guide OSA clinical decision making, without much success [6].

Peroxiredoxins (Prxs) constitute a group of redox enzymes that eliminate hydrogen peroxide using thioredoxin as the substrate; Prx1–6 isoforms have been identified [7]. Of the Prx antioxidative proteins, Prx2 has become a research focus of attention as a possible marker of oxidative stress [8,9]. Hyperoxidized Prx2 (Prx2-SO_2/3_) can serve as an indicator of oxidation during blood preservation [10,11,12]. Additionally, hyperoxidized forms of the Prx family of proteins have been found in neuronal cells [13], in red blood cells (RBCs) of patients with Alzheimer’s disease [14], and in peripheral blood mononuclear cells of patients with asthma [15]. However, the oxidation state of Prx2 in various diseases and its possible use as an oxidative stress marker in human RBCs are still being investigated.

We investigated the oxidative process of Prx2 using human RBCs [16,17,18]. To characterize Prx2-SO_2/3_, we previously developed a novel method to separate Prx2-SO_2/3_ in human RBCs using reverse-phase high-performance liquid chromatography (HPLC) and liquid chromatography-tandem mass spectrometry (LC-MS/MS) [16,17]. Moreover, our recent study has indicated that irreversible hyperoxidation of the Prx2 monomer in RBCs is easily caused by organic hydroperoxide, and that it is important to detect the hyperoxidation of Prx2 because Prx2-SO_2/3_ is a potential marker of oxidative injury in human RBCs [18]. Therefore, we expected that the hyperoxidized forms of Prx2 in RBCs might serve as biomarkers of OSA, in which oxidative stress plays an important role.

## 2. Materials and Methods

### 2.1. Chemicals

RIPA buffer was purchased from Sigma-Aldrich Corp. (St. Louis, MO, USA). Dithiothreitol (DTT) and stripping buffer were obtained from Fujifilm Wako (Osaka, Japan). All the other reagents used were of the highest commercially available grade.

### 2.2. Participants and Ethical Considerations

The age-matched control subjects (*n* = 32) comprised 32 men (mean age, 56.1 years). OSA patients (*n* = 32) comprised 32 men (mean age, 57.4 years). Both sets of patients were recruited from Fukujuji Hospital. The local ethics board approved the study protocol, and all patients provided written informed consent before participation. OSA was defined as an obstructive apnea-hypopnea index (AHI) > 5 events/h) and was diagnosed using polysomnography. OSA patients were classified into three groups according to AHI: mild (5–15, *n*= 9), moderate (15–30, *n* = 10), and severe (> 30, *n* = 13). Exclusion criteria were heart disease, brain disease, pulmonary disease, kidney disease, liver disease, and carcinoma. The study was approved by the Ethics Review Committee of Meiji Pharmaceutical University (approval number: 201967), Shonan University of Medical Sciences (approval number: 21-030), and Fukujuji Hospital (approval number: 22008), and was conducted in accordance with the principles of the Declaration of Helsinki.

### 2.3. Preparation of RBC Lysates

Protein lysates from human RBCs were prepared, as previously described [16], with modifications. Total proteins from RBCs were extracted by lysis using nine times the amount of RIPA buffer containing 5 mM of DTT. After centrifugation at 12,000× *g* at 4 °C for 5 min, the supernatants were further diluted 20-fold and used as samples for Western blotting. For chromatographic analysis using reverse-phase HPLC system (Hitachi, Ltd., Japan), the supernatant of 10-fold diluted hemolysate was reacted with an equal volume of 0.1 M Tris-HCl buffer containing 100 mM *N*-ethylmaleimide (NEM) at 37 °C for 15 min. A total of 20 μl of the reaction mixture was subjected to HPLC. The protein concentrations of the supernatants were measured after appropriate dilution using the Pierce BCA Protein Assay Kit (Thermo Fisher Scientific Inc., Japan).

### 2.4. Western Blotting and Prx2 and Prx2-SO_2/3_ Detection

Equal amounts of protein were subjected to SDS-PAGE using 12.5% or 5–20% gradient gels. For the immunoblots of Prx2 and Prx2-SO_2/3_, the proteins were transferred to a polyvinylidene difluoride membrane. The membranes were blocked with Block Ace (Yukijirushi, Tokyo, Japan) and washed three times with phosphate buffered saline containing 0.1% Tween (*v*/*v*). The membranes were incubated for 1 h at room temperature with a primary monoclonal anti-Prx2 antibody (Abnova Co., Taipei, Taiwan) or primary polyclonal anti-Prx-SO_2/3_ antibody (Abcam Co., Cambridge, UK). After washing, the blots were incubated at room temperature for 1 h with anti-mouse or anti-rabbit IgG horseradish peroxidase-conjugated secondary antibody (Vector Laboratories, Burlingame, CA, USA). Protein bands were detected and densitometrically analyzed using Luminata™ Crescendo Western horseradish peroxidase substrate (EMD Millipore, Billerica, MA, USA) in a ChemiDoc Touch Imaging System (Bio-Rad Laboratories, Tokyo, Japan). The membrane was treated with stripping buffer, according to the manufacturer’s instructions, and then re-probed. The band analysis tools of ImageLab software version 4.1 (Bio-Rad Laboratories, Tokyo, Japan) were used to select and determine the background-subtracted density of the bands in all blots.

### 2.5. Separation and Fractionation of Reduced Prx2 and Prx2-SO_2/3_ by Reverse-Phase HPLC

The RBC hemolysate (10-fold dilution) treated with NEM was separated by reverse-phase HPLC, as previously described [16]. Briefly, the RBC lysate was injected into an HPLC column (YMC-packed PROTEIN-RP; YMC Co., Tokyo, Japan) equipped with an ultraviolet detector (280 nm). Flow phase A was water containing 0.1% (*v*/*v*) trifluoroacetic acid. Flow phase B consisted of acetonitrile containing 0.1% (v/v) trifluoroacetic acid. The gradient program was as follows: 40% B (0 min), 40% B (20 min), and 45% B (50 min). The eluted fractions corresponding to 29–45 min (1 mL/min; 16 fractions) were collected in tubes and concentrated to dryness to prepare the samples for Western blotting.

### 2.6. Statistical Analysis

Statistical evaluations were performed using Welch’s *t*-test (R version 4.0.0). Values are presented as the mean ± standard deviation. Differences with *p* < 0.05 were considered statistically significant. Pearson correlation coefficients were used to statistically evaluate correlations between the Prx2-SO_2/3_ level (band intensity) in RBCs and AHI, age, and BMI, respectively.

## 3. Results

Samples were collected consistently from patients with mild to severe OSA to estimate Prx2-SO_2/3_ as a biomarker of OSA. Age- and sex-matched healthy subjects were enrolled as controls. As shown in Table 1, no significant differences in sex ratio or age were observed between the healthy subjects and patients. Prx2-SO_2/3_ accumulation in RBCs was obviously increased in patients with mild, moderate, and severe OSA (total *n* = 6; Figure 1A). Densitometric analysis revealed that the band intensity (Prx2-SO_2/3_/Prx2) was more than 10-fold higher than that in the corresponding healthy subjects (*n* = 6; Figure 1B). Based on these results, we further examined Prx2-SO_2/3_ accumulation in the RBCs of more patients with OSA (*n* = 26) and healthy subjects (*n* = 26). A significant difference in band intensities was clearly observed between patients with OSA (*n* = 32) and healthy subjects (*n* = 32), while there was a relatively even variation in the patient population (Figure 2). Thus, to determine the factors contributing to differences in the accumulation of Prx2-SO_2/3_ in patients with OSA, we examined the correlation between band intensity (Prx2-SO_2/3_/Prx2) and AHI, age, and BMI. The results showed a clear positive correlation between band intensity and AHI (Figure 3A). In contrast, there was a weak correlation between band intensity and age, and no correlation with BMI (Figure 3B,C). Moreover, to analyze the oxidation state of Prx2 in the RBCs of OSA patients, we separated the lysates of the RBCs from healthy subjects and patients with OSA (*n* = 3, each) by reverse-phase HPLC and analyzed the elution fractions (30–45 min, 1 mL/min) by Western blotting using anti-Prx2 and anti-Prx-SO_2/3_ antibodies. Representative results for each group are shown in Figure 4A,B. In healthy subject (control), elution of reduced Prx2 was observed between 39–43 min (Figure 4A), whereas multiple fractions containing Prx2 were eluted between 36–41 min, earlier than the reduced Prx2 in the sample from severe OSA patients (Figure 4B). These differences in the elution patterns from healthy subjects were commonly observed in the three samples from patients with OSA. Moreover, Western blotting results revealed that the Prx2 in all fractions eluted between 35–39 min in the chromatogram of OSA patients were hyperoxidized.

## 4. Discussion

OSA is a typical pathological condition characterized by increased oxidative stress. Studies have measured and compared levels of thiobarbituric acid reactive substances in the blood [19,20] and 8-hydroxyguanosine in the urine [21,22] as biomarkers for OSA. However, no unified opinion currently exists on the changes in their levels, and an appropriate and practical diagnostic marker has yet to be discovered for OSA.

We propose that Prx2 may serve as a biomarker for oxidative stress diseases, since it reacts with various reactive oxygen species in RBCs to produce a dimer via disulfide bonds, as well as its hyperoxidized form (Prx2-SO_2/3_) [16,17,18]. Therefore, in this study, we investigated the oxidation state of Prx2 in patients with hypoxic stress-induced respiratory diseases using our established method for analyzing Prx2-SO_2/3_ and compared it with that in healthy subjects to examine its potential as a biomarker.

The Prx2 oxidation state may serve as a biomarker under several conditions. Yoshida et al. found that RBCs from patients with Alzheimer’s disease have elevated levels of oxidized Prx2, indicating that it is a potential biomarker for the early detection of Alzheimer’s disease [14]. Bayer et al. showed that leukocyte-mediated oxidation of erythrocytic Prx2 could be a marker of inflammation [23]. Kwon et al. reported hyperoxidized Prx1, 2, 3, and 6 accumulation in peripheral blood monocytes of patients with severe asthma [15]. We could not compare our results for Prx1 and Prx3 with the levels reported by Kwon et al. because Prx1 and Prx3 were nearly absent in the RBCs used in this study. However, Prx6 levels did not change significantly. Recently, Penque et al. reported that Prx2 in the RBCs of patients with OSA is oxidized, and that Prx2 dimers or multimers may serve as diagnostic markers for OSA [24]. However, since significant Prx2-SO_2/3_ accumulation was unclear in their study, it was difficult to simply compare their results with ours.

In this study, we hypothesized that the Prx2 oxidation state in human RBCs could serve as a diagnostic marker that reflects the pathological condition of hypoxia-related diseases and examined its potential as a biomarker using clinical samples. Focusing on OSA, a hypoxia-related disease, we first measured and compared the Prx2-SO_2/3_ levels in the RBC of a few patients (*n* = 6) and age-matched them to healthy subjects (*n* = 6), which revealed an exceedingly clear difference in patients with OSA. Thus, we suspected that Prx2-SO_2/3_ was abundantly produced and accumulated in patients with OSA. Based on the results of these patients, we further collected RBC samples and measured Prx2-SO_2/3_ levels in RBCs from a larger number of patients and age-matched healthy subjects (*n* = 32 each). As the first study using a clinical sample, we decided to conduct a comparative experiment with about 10 patients each with mild, moderate, and severe OSA, for a total of more than 30 patients. As shown in Figure 2, the results of the comparison strongly suggest that the accumulation of Prx2-SO_2/3_ is a specific oxidative status observed in patients with OSA because erythrocytic Prx2-SO_2/3_ levels in patients with OSA (AHI > 5) were significantly higher than those in healthy subjects. However, a relatively large variation in the hyperoxidation state of Prx2 was observed in the OSA patient group (Figure 2). The reason for this variation was likely that the 32 samples were obtained from patients with mild (5 < AHI < 15), moderate (15 < AHI < 30), and severe (AHI > 30) OSA conditions. Therefore, we investigated the relationship between the band intensity of Prx2-SO_2/3_ and the severity of the symptoms. A clear positive correlation was evident between the hyperoxidation state of Prx2 and the level of OSA symptoms (Figure 3A). This result suggests that Prx2-SO_2/3_ accumulation may be an indicator of the severity of OSA. It is likely that age and BMI are positively associated with AHI. Our results indicated that the age of patient may be involved in the accumulation of Prx2-SO_2/3_, albeit weakly (Figure 3B). However, it seems that the oxidation state of Prx2 is independent of the BMI in patients with OSA in this study (Figure 3C).

We previously established a method for the separation and detection of Prx2-SO_2/3_ and reduced Prx2 [18,25]. Our method revealed that more hyperoxidized Prx2-SO_3_ eluted faster than reduced Prx2. In the present study, the method was applied to samples from OSA patients and healthy subjects to compare the separation patterns and detect Prx2 and Prx2-SO_2/3_. The present results suggest that Prx2-SO_2/3_ accumulation in the RBCs of patients with OSA is likely Prx2-SO_3_ because multiple hyperoxidized Prx2 molecules from these patients displayed faster elution times than those of the control samples in reverse-phase HPLC (Figure 4). In the future, it is desirable to clarify the oxidation site of Prx2 in RBCs of patients with OSA by identifying the sulfonic acid groups in the molecule using the LC-MS/MS method [25].

## 5. Conclusions

Significantly higher levels of Prx2 hyperoxidation were observed in patients with OSA compared to those in healthy subjects. A significant correlation was evident between the severity of OSA and the levels of Prx2 hyperoxidation. These findings suggest that Prx2-SO_2/3_ is a promising diagnostic marker candidate for OSA. In recent years, it has become clear that periodic systemic hypoxia due to OSA is an independent risk factor for cardiovascular complications, as it produces excess reactive oxygen species, which causes cell damage in various organs [26]. In clinical practice, early OSA detection is crucial for avoiding cardiac and cerebrovascular disease risk, and research on diagnostic markers for OSA has been attracting significant attention [21]. Hence, further investigation is required to demonstrate the practical usefulness of our findings.

## Figures and Tables

**Figure 1 antioxidants-11-02486-f001:**
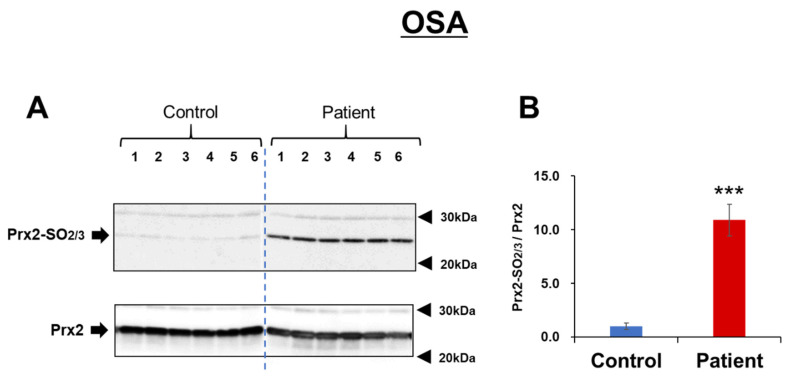
Hyperoxidation of Prx2 in RBC from OSA patients. (**A**): Samples were prepared as described in the Materials and Methods section. Each hemolysate from six patients with OSA (right side; lanes 1, 2: mild; lanes 3, 4: moderate; lanes 5, 6: severe) or six healthy subjects (left side; lanes 1–6) were analyzed by Western blotting with anti-Prx-SO_2/3_ antibody (upper image) or anti-Prx2 antibody (lower image) followed with 12.5% SDS-PAGE. (**B**): The density of the Prx2-SO_2/3_ bands was measured and normalized to the density of Prx2. The band intensity of the patient group (*n* = 6) was expressed as a fold-change relative to the control group (*n* = 6). *** *p* < 0.001 versus control group.

**Figure 2 antioxidants-11-02486-f002:**
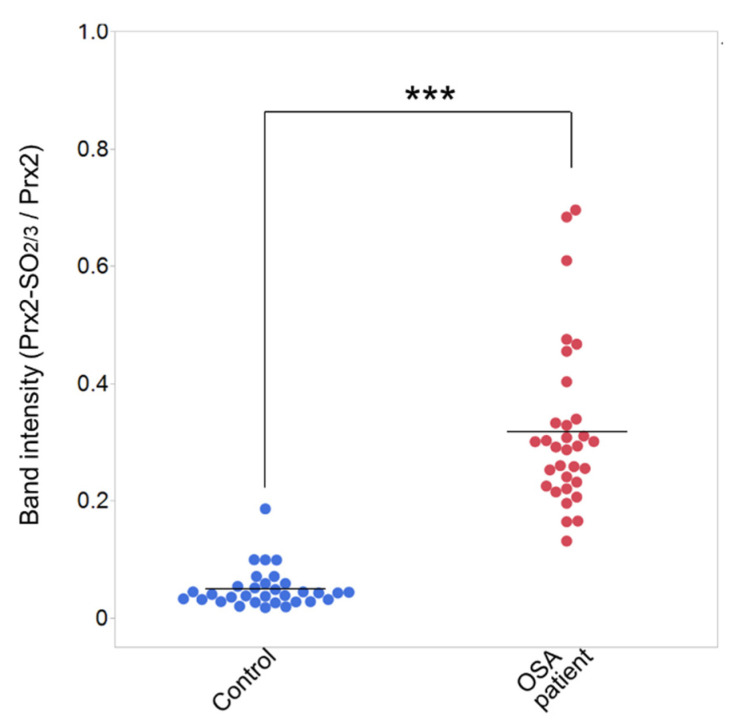
Beeswarm plots for band intensities (Prx2-SO_2/3_/Prx2) of healthy subjects and OSA patients. The densities of Prx2-SO_2/3_ bands were measured and normalized to that of Prx2. The band intensity of patients with OSA (*n* = 32) and healthy control subjects (*n* = 32) was expressed as the normalized ratio of density (Prx2-SO_2/3_/Prx2). *** *p* < 0.001 versus control group.

**Figure 3 antioxidants-11-02486-f003:**
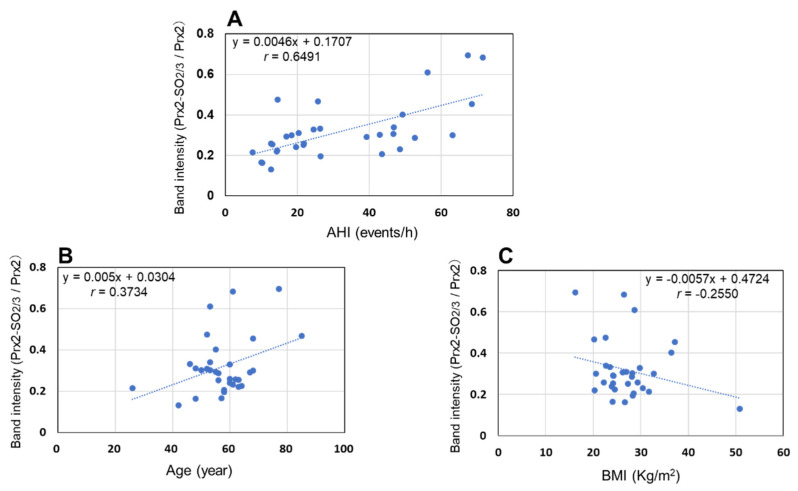
Correlation between the band intensities of Prx2-SO_2/3_ and AHI, age, and BMI corresponding to each patient with OSA. The band intensity of patients with OSA (*n* = 32) was expressed as the normalized ratio of density (Prx2-SO_2/3_/Prx2). AHI was measured using a polysomnography test. BMI was calculated using a person’s height and weight (kg/m^2^). (**A**): correlation between band intensity and AHI; (**B**): correlation between band intensity and age; (**C**): correlation between band intensity and BMI.

**Figure 4 antioxidants-11-02486-f004:**
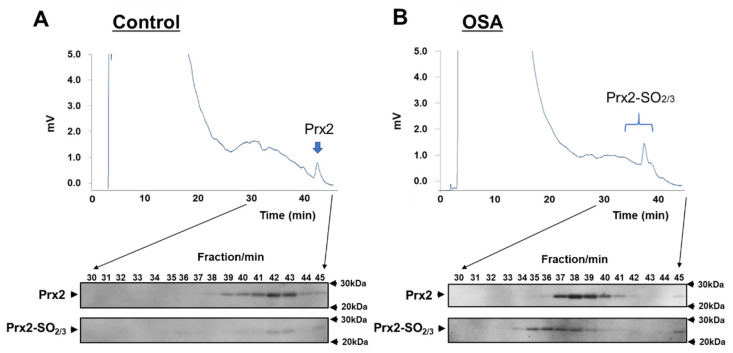
Chromatographic analysis of Prx2 oxidation in RBC lysates from a healthy subject and a patient with OSA. The samples were prepared as described in the Materials and Methods section. Each representative result from the OSA patient group (*n* = 3) and control group (*n* = 3), respectively, was shown. Chromatograms of RBC lysates from a healthy control (**A**) and a patient with OSA (**B**). Proteins in the 16 fractions (29–45 min) were collected and concentrated. Each dried sample was resolved in SDS buffer and subjected to 5–20% SDS-PAGE, followed by Western blotting with anti-Prx2 antibody (upper image). The same PVDF membranes were stripped and re-probed with anti-Prx2-SO_2/3_ antibody (lower image).

**Table 1 antioxidants-11-02486-t001:** Subject characteristics.

Parameters	Healthy Subjects	OSA Patients
Number of subjects	32	32
Age (years)	56.1 ± 5.39	57.4 ± 10.0
Sex (M/F)	32/0	32/0
BMI (kg/m^2^)	22.9 ± 4.06	27.1 ± 6.07
AHI (events/h)	―	32.1 ± 19.1

BMI: body mass index; AHI: apnea hypopnea index.

## Data Availability

The data are contained within the article.

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
