# Peer review of "Hyperoxidized Peroxiredoxin 2 Is a Possible Biomarker for the Diagnosis of Obstructive Sleep Apnea"

_antioxidants, 2022, doi:10.3390/antiox11122486_

Round 1
Reviewer 1 Report
Dear Authors,
I am asked to review this manuscript for you. Although this study is interesting, there are some major issues that should be addressed before further consideration. Please consider my suggestions.
1. This study is a preliminary study. Therefore, the title should be more conservative.
2. The study included two different disorders of the airway (COPD: lower airway; OSA: upper airway) with distinct pathophysiological mechanisms. Unfortunately, the relatively small patient number in the COPD group makes it hard to make any conclusion and contributes to the statistical analysis unprecise. Therefore, I suggest removing all the parts related to COPD to make this manuscript more concise.
3. This study included three sites with different approval numbers. Did these IRB approvals permit the same study?
4. Lines 80-84 were very confusing. The study flow chart should be added to the Materials and Methods to make it clear. Furthermore, the participant's data should be summarized in the Result. Moreover, what were the controls and patients matched for? Were the confounding factors of increased oxidative stress matched?
5. The effects of age and BMI on the relationship between OSA severity and band intensity were not clear. In patients with OSA, age and BMI are positively associated with AHI. How did the authors exclude the effect of age?
6. Normality of the continuous variables should be determined, then choose correct statistical methods.
7. Figure 1 was missing.
8. Results need to be rearranged.
9. The authors should enhance the technical novelty of determining Prx2-SO2/3 for OSA since these experiments detecting Prx2-SO2/3 have been established in their previous studies.
Author Response
- This study is a preliminary study. Therefore, the title should be more conservative.
According to reviewer’s suggestion, we changed the title conservatively.
Please see the new title of revised MS.
- The study included two different disorders of the airway (COPD: lower airway; OSA: upper airway) with distinct pathophysiological mechanisms. Unfortunately, the relatively small patient number in the COPD group makes it hard to make any conclusion and contributes to the statistical analysis unprecise. Therefore, I suggest removing all the parts related to COPD to make this manuscript more concise.
According to reviewer’s suggestion, we removed all the parts related to COPD in the revised MS.
- This study included three sites with different approval numbers. Did these IRB approvals permit the same study?
This study was conducted in collaboration with three organizations. Therefore, three approval numbers are listed because we have obtained approval from our respective IRBs for common research project.
- Lines 80-84 were very confusing. The study flow chart should be added to the Materials and Methods to make it clear. Furthermore, the participant's data should be summarized in the Result. Moreover, what were the controls and patients matched for? Were the confounding factors of increased oxidative stress matched?
As the reviewer pointed out, the relevant part (Lines 80-84 in the previous MS) of “Participant and ethical consideration” was not clear, therefore we have rewritten it in the revised MS.
Page 2, line74 -75
Although a flowchart is not shown because the description was simplified by excluding COPD and including only patients with OSA, the data covered are summarized in new Table 1 in the "Results" section of the revised MS.
Please see Table 1 in the revised MS
Also, controls and OSA patients matched for age and gender. We think that the confounding factors of increased oxidative stress were not matched.
Page 3, line 131-135
- The effects of age and BMI on the relationship between OSA severity and band intensity were not clear. In patients with OSA, age and BMI are positively associated with AHI. How did the authors exclude the effect of age?
Considering the reviewer’s comments, in addition to correlations with AHI, we newly investigated correlations between accumulated Prx2-SO2/3 (band intensity) and age or BMI of OSA patients. The results showed a weak correlation with age, but no correlation with BMI in the OSA patient samples used in this study. Therefore, it is possible the band intensity (oxidation state) of Prx2-SO2/3 is associated with age of OSA patient. However, since the ages of the OSA patients collected in this study are close and not equally distributed, we thought it difficult to clearly state the correlation between Prx2-SO2/3 and age of OSA patient.
We consider that no positive correlation between the band intensity of Prx-SO2/3 and BMI is due to the small number of patients with OSA who are extremely obese in the present study. Also, we observed no correlation between AHI and BMI in this study (Data is not shown).
Please see the results of the new correlations shown in Fig. 3B and C of the revised MS.
Page 3, line 141-146 in “Results” section
Page 7, line 238-241 in “Discussion Section”
- Normality of the continuous variables should be determined, then choose correct statistical methods.
According to reviewer’s suggestion, we have reconsidered the statistical analysis.
In the revised MS, we performed statistical analysis of the normality and variance of the data shown in Fig. 1 and Fig. 2. As a result, we found that both data were not normally distributed and also had unequal variances, so the Welch-test was adopted as a statistical method for testing significant differences between the two groups, and the recalculated results are shown in Fig.1 and 2 in the revised MS.
Please see page 3, line 125
- Figure 1 was missing.
Fig. 1, which showed COPD data, has been deleted in the revised MS.
- Results need to be rearranged.
According to reviewer’s suggestion, we have rearranged the results using only OSAS data with subject characteristics (Table 1) and new correlation results (Fig. 3B, C).
- The authors should enhance the technical novelty of determining Prx2-SO2/3 for OSA since these experiments detecting Prx2-SO2/3 have been established in their previous studies.
As the reviewer pointed out, I agree that further technical novelty in the analysis of the oxidation state of Prx2 would be desirable. However, we present the results of the first clinical application using our established advanced methods in this report. Accordingly, we have added sentences in the “Discussion” section to explain the background for this clinical application, detailing the results obtained with the method we developed.
Page 7, line 242-246.
Moreover, we mentioned that we will attempt to analyze the oxidation sites of Prx2 in OSA patients using a new method based on LC-MS/MS in the Discussion section.
Please see page 7, line 249-250
Thank you very much for reviewer’s comments with significant suggestions.
We expect that these improvements altered in this new MS make our article clearer and more interesting.
Reviewer 2 Report
The authors investigated that the hyperoxidized forms of Prx2 in RBCs might be as a novel biomarker for the diagnosis of OSAS.
<Major Comments>
Materials and Methods:
1. This study was conducted as a prospective study by matching normal controls to patients with COPD and OSA. I am curious about the calculation method for determining the number of target patients. A description of how to determine the number of target patients is required.
2. It would be better to have a table of the clinical characteristics of the disease groups participating in this study and the matched normal control group.
Results:
3. Figure 1 was not shown in the manuscript.
4. Why did you use data from the only 6 patients with severe OSA among 13 in Figures 2A and 2B? For PRx2, it is necessary to present your results regarding comparison between OSA patients and matched normal controls.
5. Figure 5 shows the results of comparing OSA and the normal control group. Were the results for the normal control group confirmed by pooled samples from 32 people? Also, why did only one person's sample be used for comparison with the normal control group in patients with severe OSA?
Discussion:
6. No significant results were found in patients with COPD in this study. You have explained the reason for this is the small number of subjects and the advanced age of the patients. Do older people show different trends in Prx2 levels?
Could it be that the severity of the COPD patient was not severe, that is, the patient was not exposed to hypoxia?
Author Response
<Major Comments>
Materials and Methods:
- This study was conducted as a prospective study by matching normal controls to patients with COPD and OSA. I am curious about the calculation method for determining the number of target patients. A description of how to determine the number of target patients is required.
We do not know that there is any rule for determining the number of patients, however, the evaluation with 6 COPD patients is insufficient and has been removed from this article.
However, in this pioneering study, which was the first to test the potential of Prx2-SO2/3 as a biomarker in OSAS using patient samples, we determined that the number of samples needed for evaluation by statistical analysis was more than 30. In fact, many other papers have evaluated biomarkers with as few as 30 people (an example for the paper cited in this article as [22]).
Considering the reviewer’s comments, the reason for the decision on the number of patient samples was mentioned in the Discussion section.
Please see page7, line 222-228
- It would be better to have a table of the clinical characteristics of the disease groups participating in this study and the matched normal control group.
According to reviewer’s suggestion, the data covered are summarized in Table 1 in the "Results" section of the revised MS.
Please see new Table 1
Results:
- Figure 1 was not shown in the manuscript.
In the previous MS, Fig.1 was missing due to some trouble. In the revised MS, the results showing COPD patients (Fig. 1 in the previous MS) have been removed.
- Why did you use data from the only 6 patients with severe OSA among 13 in Figures 2A and 2B? For PRx2, it is necessary to present your results regarding comparison between OSA patients and matched normal controls.
In Fig.2 (in the previous MS), the statement that all six samples obtained from patients had severe OSAS was incorrect. In Fig.2A, B shown in the previous MS (Fig.1A, B in the revised MS), the result of Western blot (total 12 lanes) contains two severe (AHI>30), two moderate (15<AHI<30), and two mild (5<AHI<15) patients (right side), and six healthy subjects (left side). Thus, we think this result of comparison between OSA patients and normal controls clearly shows the difference of the accumulation of hyperoxidized prx2 (Prx2-SO2/3).
The severity of the patients in the six samples was correctly rewritten in the “Results” section and the caption of Fig.1 in the revised MS.
Please see page 3, lane 134-135, and page 4, line 163-164
Although there is no difference in the amount of Prx2 itself (oxidized + reduced form) between patients and normal subjects, the difference in band intensity ratio (hyperoxidized Prx2/total Prx2) is visually evident in Fig. 1A (in the revised MS) and quantified in Fig. 1B (in the revised MS) shows a difference of more than 10-fold. We considered 6 samples of each group would be sufficient to show the differences in the data of western blot for visual clarity.
Moreover, the band intensity ratio (Prx-SO2/3/Prx2) of healthy subjects (Control) converged, whereas the band intensity of Prx2-SO2/3 was apparently much stronger in OSA patients than in Control, as was the case for the 26 OSA samples measured after the 6 samples (Fig.1 in the revised MS) examined first. However, when density was measured and the ratio of oxidized forms (Prx-SO2/3/Prx2) was quantified, considerable variation was observed in OSA patients compared to healthy subjects, as shown in Fig. 2 (in the revised MS).
Please see Fig.1 and Fig.2 (revised MS).
Page 3, line 135-141
- Figure 5 shows the results of comparing OSA and the normal control group. Were the results for the normal control group confirmed by pooled samples from 32 people? Also, why did only one person's sample be used for comparison with the normal control group in patients with severe OSA?
The separation pattern shown in Fig. 4A (revised MS) represents a result of a sample from a healthy subject. Also shown in 4B is the result of a severe OSA patient (AHI: 56.2).
Considering the reviewer’s comments, we performed additional experiments. When the same analysis was performed on two healthy subjects and two OSA patients (AHI: 27.4 and 14.5), Prx2 eluted with the same retention time in the healthy subjects, while the higher the patient's AHI, the more Prx2 in the erythrocytes was hyperoxidized and eluted faster from the reversed-phase column (attached Fig.A).
Therefore, Figure 4 shows a representative pattern with clear differences in one case each of a healthy subject and a patient with severe OSA.
Please see the HPLC separation patterns of a group of healthy subjects and a group of OSA patients, each consisting of three samples, including the results of an additional experiment shown in Fig. A.
To avoid any misunderstanding that this is a specific case, a note has been added to the "Results" section and to the figure captions stating that these are representative results obtained by measuring three samples from each group.
Page 3, line 147– line 150
Discussion:
- No significant results were found in patients with COPD in this study. You have explained the reason for this is the small number of subjects and the advanced age of the patients. Do older people show different trends in Prx2 levels?
Despite the small number of samples, a significant increase (1.6-fold) in the ratio of hyperoxidized Prx2 in RBCs was observed in COPD patients compared to that in healthy subjects. However, However, we have removed all sections on COPD in this paper as we consider the data insufficient.
The oxidation of Prx2 in human erythrocytes during aging is not yet clear, but it is known that oxidation is enhanced during storage in blood for transfusion, as cited in this MS.
Could it be that the severity of the COPD patient was not severe, that is, the patient was not exposed to hypoxia?
In patients with COPD, we would expect to observe hypoxia even in less severe cases. I would be happy to refer you to GOLD (Global Initiative for Chronic Obstructive Lung Disease) for more information on the degree of hypoxia in COPD patients and its relationship to the severity of the disease.
We will not mention COPD at this time as we have removed all content related to COPD, but we will keep your question in mind in future studies.
Thank you very much for reviewer’s comments with significant suggestions.
We expect that these improvements altered in this new MS make our article clearer and more interesting.

Round 2
Reviewer 1 Report
Dear authors,
I admire you because you addressed all my previous issues in the modified manuscript. I have no more questions. Two minor comments are further provided. (1) Please keep the same using OSAS or OSA in the title, abstract, and text. (2) Please provide a correct definition of OSAS. Obstructive AHI is more suitable than AHI to define OSAS. Accordingly, I recommend the editors consider accepting this manuscript after minor revisions. Congratulations.
Author Response
Reviewer 1:
Dear Authors,
I admire you because you addressed all my previous issues in the modified manuscript. I have no more questions. Two minor comments are further provided. (1) Please keep the same using OSAS or OSA in the title, abstract, and text. (2) Please provide a correct definition of OSAS. Obstructive AHI is more suitable than AHI to define OSAS. Accordingly, I recommend the editors consider accepting this manuscript after minor revisions.
- Please keep the same using OSAS or OSA in the title, abstract, and text.
According to reviewer’s suggestion, OSA was used in this MS to provide consistency.
Please see the 2nd revised MS.
- Please provide a correct definition of OSAS. Obstructive AHI is more suitable than AHI to define OSAS.
According to reviewer’s suggestion, we have provided the correct definition of OSA based on the cited guidelines (Ref. [2]) in the "Introduction section".
Please see page 1, lines 38 - 41 in the 2nd revised MS.
Also, "AHI" is used in this paper to mean the same thing as "obstructive AHI". We have added the phrase "Obstructive AHI" where it first appeared in the M&M section. However, since very few papers use the term "obstructive AHI," we have used the term "AHI" in subsequent texts.
Please see page 2, line 78 in the 2nd revised MS.
Thank you very much for reviewer’s comments with significant suggestions.
We expect that these improvements altered in this 2nd revised MS make our article clearer and more interesting.
Reviewer 2 Report
The authors provided detailed explanations and appropriate corrections to the comments.
Author Response
Reviewer 2
The authors provided detailed explanations and appropriate corrections to the comments.
Thank you very much for the reviewer’s response.
We sincerely thank the reviewer for the thoughtful comments.